# Prevalence of *Neospora caninum* and *Toxoplasma gondii* Antibodies and DNA in Raw Milk of Various Ruminants in Egypt

**DOI:** 10.3390/pathogens11111305

**Published:** 2022-11-07

**Authors:** Ragab M. Fereig, Hanan H. Abdelbaky, Amira M. Mazeed, El-Sayed El-Alfy, Somaya Saleh, Mosaab A. Omar, Abdullah F. Alsayeqh, Caroline F. Frey

**Affiliations:** 1Division of Internal Medicine, Department of Animal Medicine, Faculty of Veterinary Medicine, South Valley University, Qena 83523, Egypt; 2Veterinary Clinic, Veterinary Directorate, Qena 83511, Egypt; 3Department of Infectious Disease, Faculty of Veterinary Medicine, Arish University, North Sinai, Arish 45516, Egypt; 4Department of Parasitology, Faculty of Veterinary Medicine, Mansoura University, Mansoura 35516, Egypt; 5Department of Parasitology, Faculty of Veterinary Medicine, South Valley University, Qena 83523, Egypt; 6Department of Veterinary Medicine, College of Agriculture and Veterinary Medicine, Qassim University, Buraidah 51452, Qassim, Saudi Arabia; 7Department of Infectious Diseases and Pathobiology, Institute of Parasitology, Vetsuisse Faculty, University of Bern, Länggassstrasse 122, CH-3012 Bern, Switzerland

**Keywords:** cattle, ELISA, neosporosis, PCR, sheep, toxoplasmosis

## Abstract

The prevalence of *Neospora caninum* and *Toxoplasma gondii* antibodies in raw milk samples was estimated in different ruminants and Egyptian governorates. Of 13 bulk milk samples tested by ELISA, five (38.5%) were positive for antibodies to *N. caninum*, and two samples were additionally positive for antibodies to *T. gondii*, resulting in a seroprevalence of 15.4% for both *T. gondii* and co-infection. In individual milk samples (n = 171) from the same bulks, antibodies to *N. caninum* were detected in 25.7%, to *T. gondii* in 14%, and 3.5% had antibodies to both parasites. A strong correlation between the OD values of the bulk samples and of the relevant individual milk samples was found for *T. gondii* (Pearson r = 0.9759) and moderately strong for *N. caninum* (Pearson r = 0.5801). Risk factor assessment for individual milk samples revealed that antibodies to *T. gondii* were significantly influenced by animal species, while no risk factors were detected for *N. caninum* antibodies. Additionally, DNA of *N. caninum* was detected in a bulk milk sample of cattle for the first time in Egypt, and DNA of *T. gondii* was found in bulk milk samples of cattle, sheep and goats. This is the first study in Egypt in which bulk milk samples of different ruminants were tested for the presence of *N. caninum* and *T. gondii* antibodies and DNA. Both individual and bulk milk samples are useful tools for monitoring antibody response to *N. caninum* and *T. gondii* infections in different ruminants in Egypt.

## 1. Introduction

*Neospora caninum* is a protozoan parasite that causes sporadic, endemic and epidemic abortions in cattle worldwide [1,2]. Endogenous transplancental transmission from dam to calf during pregnancy appears to be the main route of transmission [1,3]. Horizontal transmission of cattle by ingestion of sporulated oocysts shed by canids as final hosts can also occur [4,5]. *Neospora caninum* causes reproductive diseases which have a negative economic impact not only on milk yield but also on the replacement of dairy cows [6,7].

Milk has been demonstrated as a suitable sample for the detection of specific antibodies to *N. caninum* [8]. The antibodies found in milk, primarily IgG, are selectively transported from the serum into the mammary gland [9]. Milk samples from seropositive cows showed a significantly higher IgG level when compared with those from seronegative cows [10,11]. A high agreement, namely 95%, was recorded in antibody response between serum and milk [12]. Additionally, an excellent agreement among different diagnostic techniques in serum and milk samples was obtained [11,13]. Consistently, milk samples from sheep have been evaluated and validated for the detection of anti-*N. caninum* antibodies using bulk or individual samples [14,15,16].

Toxoplasmosis is an important and prevalent foodborne parasitic disease caused by *Toxoplasma gondii*. This protozoon can infect almost all warm-blooded animals, including human beings and farm animals [17]. The infection can occur in three ways: congenital transmission, organ transplant/blood transfusion, and contaminated food and water [18]. Human infection occurs frequently by consumption of raw or undercooked meat containing tissue cysts, unwashed fruit and vegetables contaminated with oocysts, and potentially also by tachyzoites shed in milk [19].

*Topxoplasma gondii* antibodies have been detected in the milk of various hosts like sheep, cattle, buffaloes, and camels [20,21], goats [21,22], and even in lactating women [23]. High agreement between ELISA results obtained with serum and milk samples of the same individuals was reported [22]. 

In cases of toxoplasmosis or neosporosis, substantial economic losses have been attributable directly to abortions in farm animals. Not only meat but also dairy sheep farms in Spain have suffered from economic losses resulting from *T. gondii* abortions [24]. Consistently, although neosporosis has been reported as a common cause of abortion in cattle, it has also been detected as a cause of abortion and perinatal mortality in sheep and goats [25,26,27]. Regarding public health, specific antibodies to *N. caninum* have been detected in women’s sera using IgG [28] or IgM [29]. However, no evidence of a human clinical form of neosporosis has been detected yet. A recent study, however, discovered that two samples (1%) out of 201 examined human umbilical cord blood samples were Nc5 PCR-positive for *N. caninum*, while the placenta tested negative for this parasite [30]. On the contrary, approximately 25% of all people have experienced *T. gondii* infection globally. Congenital infections carry the risk of more serious results, and the growing foetus may exhibit symptoms that range from severe to moderate. Severe forms of congenital toxoplasmosis can cause congenital anomalies and even loss of life [31]. Severe infections in immunocompromised people often arise from the recurrence of a persistent illness [32].

In Egypt, antibodies to *N. caninum* and *T. gondii*, respectively, have been detected in numerous hosts using serum samples [33,34,35,36]. For milk samples, however, only one study looked for specific anti-*N. caninum* antibodies in cow’s milk in Egypt so far [37]. Anti-*T. gondii* antibodies have been assessed in Egypt in human mothers’ milk [23], donkey’s milk [38], milk of sheep, cattle, buffaloes, and camels [21], and milk of sheep, goats and cows [39]. However, risk factors for antibodies in milk and the testing of bulk milk samples have not yet been assessed. Milk samples can represent a valuable, cheap, and non-invasive tool in monitoring *N. caninum* and *T. gondii* prevalence as alternatives to serum samples [40]. This current study aims at advancing the knowledge on surveillance of these important parasites using milk samples of ruminants. 

## 2. Materials and Methods

### 2.1. Sample Collection and Preparation

Raw bulk milk samples (10 mL each) and individual raw milk samples (5 mL each) from 13 randomly chosen dairy farms and smallholders were collected from lactating cows, buffaloes, ewes and does between June to September 2022 (Table 1). Different regions were sampled in order to best represent Egypt: Dakahlia in the North, Cairo as the most densely populated governorate in the centre of Egypt, and Qena and Sohag in the South. Bulk milk samples have been collected directly at the farm (n = 7) or from shops (n = 6). During the direct visits at the farms, individual samples (n = 171) were randomly collected additionally (Table 1). Raw milk at shops usually was obtained freshly from a governmental or a private dairy farm and kept in bulk tanks for one-day use. Sometimes the milk was aliquoted in plastic bags of variable kilograms, and kept at +4 °C until selling out. No information on abortion history was available from these herds. The milk samples were centrifuged at 1000× *g* for 10 min. Lactoserum was collected from the layer below the cream layer and stored at −20 °C until used. 

### 2.2. Detection of Anti-N. caninum and Anti-T. gondii Antibodies in Milk

Samples were tested for specific antibodies to *N. caninum* using a commercially available ELISA (*Neospora caninum* Milk Competitive ELISA, ID. Vet, Grabels, France), according to the manufacturer’s instructions. Briefly, undiluted milk samples and positive and negative controls were added to the microplate and incubated at 5 °C for 20 h. On the following day, the microplate was washed thrice using the washing buffer supplemented in the commercial kit. Washing out of any milk precipitates in all wells was visually confirmed. Then, 100 µL of diluted conjugate was added to the wells and incubated at room temperature (RT) for 30 min. After three washings, 100 µL substrate was added to each well and incubated at RT in a dark place for 15 min. Finally, 100 µL stop solution was added, and the optical density (OD) was measured. The ODs obtained were used to calculate the percentage of sample (S) to negative (N) ratio (S/N%) for each of the test samples according to the following formula S/N (%) = OD sample/OD negative control × 100. Samples with an S/N% greater than 50% were considered negative and considered positive if the S/N% was less than or equal to 50%. 

Concerning *T. gondii*, the milk samples were analysed using an indirect multi-species ELISA for toxoplasmosis (ID.vet, Grabels, France) according to the manufacturer’s instructions. Milk samples were added without dilution while the controls were diluted 1:10 and tested, as reported previously [22]. The OD obtained was used to calculate the percentage of sample (S) to positive (*P*) ratio (S/P%) for each of the test samples according to the following formula: S/P (%) = (OD sample − OD negative control)/(OD positive control − OD negative control) × 100. Samples with an S/P% less than 40% were considered negative; if the S/P% was between 40% and 50%, the result was considered doubtful and considered positive if the S/P% was greater than 50%. The optical density of all ELISA results was measured at 450 nm with an Infinite^®^ F50/Robotic ELISA reader (Tecan Group Ltd., Männedorf, Switzerland).

### 2.3. DNA Extraction and Preparation 

DNA extraction from bulk milk samples was performed using the QIAamp DNA Mini kit (Qiagen, Germany, GmbH) as follows. After high-speed centrifugation (14,000 rpm/2 min) of 10 mL of milk samples, creamy and lactoserum layers were carefully discarded, and 25 mg of the pellet was mixed with 20 µL of proteinase K and 180 µL of ATL buffer and then incubated at 56 °C for 3 h. After incubation, 200 µL of AL buffer was added to the lysate, incubated for 10 min at 72 °C, and then 200 µL of 100% ethanol was added to the lysate. The lysate was then transferred to a silica column and centrifuged at 8000 rpm for 1 min. The sample was then washed and centrifuged following the manufacturer’s recommendations. Nucleic acid was eluted with 100 µL of elution buffer provided in the kit.

### 2.4. PCR Application and Analysis

References, target genes (Nc5 for *N. caninum* and B1 for *T. gondii*), primer sequences and cycling conditions for the PCRs are given in Table 2 [41,42]. Primers were supplied from Metabion International (Planegg, Germany). PCR reaction was applied in a 25 µL reaction containing 12.5 µL of EmeraldAmp Max PCR Master Mix (Takara, Shiga, Japan), 1 µL of each primer (20 pmol concentration), 5.5 µL of water, and 5 µL of DNA template. The reaction was performed in an Applied Biosystem 2720 thermal cycler (Applied Biosystems, Foster City, CA, USA). The PCR products were separated by electrophoresis on a 1.5% agarose gel (AppliChem GmbH, Darmstadt, Germany) with ethidium bromide in 1× TBE buffer at room temperature using gradients of 5 V/cm. For gel analysis, 20 µL of the products were loaded in each gel slot. Generuler 100 bp ladder (Fermentas, Hamburg, Germany) was used to determine the fragment sizes. The gel was photographed by a gel documentation system (Alpha Innotech, Biometra, San Leandro, CA, USA), and the data was analysed through computer software. The positive control DNAs were represented by field samples previously confirmed to be positive by PCR for the related genes in the Reference laboratory for veterinary quality control on poultry production, Animal health research institute, Giza, Egypt.

### 2.5. Statistical Analysis and Risk Factor Assessment

The significance of the differences in the prevalence rates and risk factor assessment was analysed with Fisher Exact Probability Test (two-tailed), 95% confidence intervals (including continuity correction) and odds ratios using an online statistical website www.vassarstats.net (accessed dates; 11–14 September 2022) as described previously [33]. *p*-values and odds ratio were also confirmed with GraphPad Prism version 5 (GraphPad Software Inc., La Jolla, CA, USA). The results were considered significant when the *p*-value was <0.05. Statistically significant differences in the OD values of the ELISA were estimated and interpreted using a *t*-test followed by the Mann–Whitney test for comparing different groups. Pearson’s correlation coefficient was applied to test the correlation between the OD values of bulk samples and the mean OD of individual milk samples from the same group. Correlation coefficients were calculated using Pearson’s correlation coefficient: |r| = 0.70, strong correlation; 0.5 < |r| < 0.7, moderately strong correlation; and |r| = 0.3–0.5 weak-to-moderate correlation [43].

## 3. Results

### 3.1. Prevalence of N. caninum and T. gondii Antibodies in Bulk and Individual Milk Samples and Detection of DNA

In this study, we investigated the prevalence of *N. caninum* and *T. gondii* antibodies in bulk and individual raw milk samples. Samples were collected from cattle, buffaloes, sheep and goats from Qena, Sohag, Cairo, and Dakahlia governorates representing different Egyptian areas. Of 13 tested bulk milk samples representing 13 different farms, positive reactions to *N. caninum* were detected in five (38.5%) samples, three of which were from cattle farms. The other two positive samples were from a sheep and a goat smallholder farm, respectively. These two latter samples also tested positive for antibodies to *T. gondii* and thus for co-infection (seroprevalence for *T. gondii* and co-infection: 15.4%) (Table 3). 

In the individual samples (n = 171) of the same bulk milk samples collected from three farms and four smallholders, antibodies to *N. caninum* were detected in 25.7%, to *T. gondii* in 14%, and 3.5% had antibodies to both parasites. In more detail, antibodies to *N. caninum* were detected in 26.2% (33/126) of the tested cattle, 18.8% (3/16) of the buffaloes, 33.3% (6/18) of the sheep, and 18.2% (2/11) of the goats, respectively (Table 4). Antibodies to *T. gondii* were detected in 81.8% (9/11) of the tested goats, 66.7% (12/18) of the sheep, 2.4% (3/126) of the cattle, and in none (0/16) of the buffaloes, respectively (Table 4). 

Detection of DNA was targeted in ELISA-positive bulk milk samples and in those containing a number of seropositive individual samples. From six bulk samples with the afore-mentioned criteria, bulk 1 from cattle showed a positive PCR result for the NC5 gene of *N. caninum* (337 bp), while bulk samples 7, 12, and 13 showed negative PCR results, although they were positive in ELISA (Figure 1). In the case of *T. gondii*, samples of bulk 12 from sheep and bulk 13 from goats that showed strong antibody reactivity using ELSA showed a positive PCR result for the B1 gene of *T. gondii* (196 bp). In addition, bulk sample 10 from cattle was also PCR-positive, despite being negative in ELISA (Figure 1). 

### 3.2. Risk Factor Assessment for N. caninum and T. gondii Infection Using Individual Milk Samples

Risk factor assessment was conducted for *T. gondii* and *N. caninum* antibodies in tested individual samples. Animal species was assessed as a risk factor for all tested samples (n = 171) collected from cattle, buffalo, sheep, and goat. Additional factors including region (Qena, Sohag, and Dakahlia), breed (native and Holstein Friesian), and management system (farm or small holder) could only be analyzed for cattle. Regarding *N. caninum*, no risk factors have been detected when analyzing any of the above-mentioned factors (Table 5). 

Regarding *T. gondii*-associated risk factors utilizing individual milk samples, the tested sheep in this study were more susceptible to *T. gondii* infection (66.7%) than cattle (2.4%; OR = 82; *p* ≤ 0.0001) and buffaloes (0%; OR = 63.5; *p* ≤ 0.0001), while goats did not differ from sheep (81.1%; OR = 0.4; *p* = 0.67) (Table 6). This was the only statistically significant association found.

### 3.3. Comparison of Antibody Responses to N. caninum and T. gondii in Individual and Bulk Milk Samples

Analysis of individual milk samples from bulk samples revealed that in negative bulk samples, positive individuals might be included, and vice versa. This effect was more pronounced in the analysis for *N. caninum* antibodies (Table 7).

Optical densities of ELISA results were visualized to compare the reactivity of positive and negative samples for both parasites to provide further validation of our testing. Strong antigen-antibody reactivity was observed in positive bulk or individual samples for both parasites, particularly in the case of *T. gondii* (Figure 2A–D). The differences between OD values of both negative and positive individual samples within the relevant bulk milk sample were analysed for both antibodies to *N. caninum* (Figure 2B) and *T. gondii* (Figure 2D), respectively.

These results were confirmed by group comparisons of antibody levels of all individual samples from negative bulks versus those from positive bulks (Figure 3A,B). Antibody levels of individual samples from bulks positive for *T. gondii* antibodies were significantly higher than that of negative bulks (*p* ≤ 0.0001) but not for *N. caninum* (*p* = 0.1810). In addition, Pearson’s correlation coefficient was used to investigate the association between the OD values of bulk samples and the mean OD values of the individual samples from the bulk. A better correlation was identified for *T. gondii* (Strong; Pearson r = 0.9759) than for *N. caninum* (Moderately strong; Pearson r = 0.5801) (Figure 3C,D). This data might be attributable to the number and distribution patterns of positive and negative individual samples among the bulk samples. 

In case of *N. caninum*, the negative bulks contained a higher percentage of positive individual samples (13%; 5/38) than what was seen for *T. gondii* (2%; 3/142). Similarly, a lower percentage of positive individual samples in positive bulks (29%; 39/133) was seen for *N. caninum*, while 72% (21/29) of the individual samples in bulks positive for *T. gondii* were positive (Figure 4). 

## 4. Discussion

The growing human population demands safe and high-quality food, including milk and dairy byproducts. The presence of *N. caninum* DNA in milk samples has been shown in a few studies for dairy cows [11,13,44] and in one study for lactating donkeys [45]. DNA of *T. gondii* was detected in milk samples from different animal species, including cattle [13], donkeys [45], as well as sheep and goats (including one report from Egypt) [21,46]. Therefore, milk might represent a potential risk of infection to suckling animals and facilitate parasite persistence in a flock. Another very important point is whether *T. gondii* in milk may be infective for humans. This risk seems to be higher in the case of consuming raw milk from an individual animal than by consumption of bulk tank milk in which the parasites would be greatly diluted [47,48].

Our study demonstrated DNA of *N. caninum* in raw milk of cattle, and of *T. gondii* in raw milk of cattle, sheep and goats. Detection of DNA in milk is correlated to the stage of parasitemia which is short and transient in case of both tested parasites [49], oppositely to the long-lasting antibody response [50,51]. While in our tested sheep and goat samples, the results of the ELISA and the PCR for *T. gondii* were both positive, the agreement between the test methods was not as nice in the cattle samples. Only one of the two tested *N. caninum* seropositive bulk milk samples showed a positive Nc5-PCR result. The cattle bulk milk sample that was positive in the B1-PCR was seronegative for *T. gondii*.

Our findings demonstrate the potential hazards of raw milk of cows, ewes or does in the transmission of *T. gondii* or *N. caninum* to susceptible hosts. Indeed, the possibility of *T. gondii* transmission via milk was estimated using experimentally infected goats. *Toxoplasma gondii* viability in goat milk and cheese was demonstrated by bioassay in cats and mice [48]. However, similar experiments have yet to be reported for *N. caninum* [44].

The current control measures for bovine neosporosis primarily rely on serological testing and the replacement of infected animals [52]. Previous results indicated that the detection of antibodies in individual milk samples is a good, non-invasive alternative option to testing serum samples for detecting anti-*N. caninum* antibodies [52,53]. However, the use of milk samples also has some limitations, as only lactating cows can be tested, and young, diseased, and dry animals are excluded from the sample [54]. Detection of neosporosis in a herd by testing a single bulk milk sample has a significant financial and logistic advantage, but sensitivity is arguably lower than when testing individual milk samples due to the diluting effect of negative milk samples in a bulk sample. The use of milk samples to detect *T. gondii* has an additional public health aspect; indeed, *T. gondii* antibodies were detected in the milk of food animals, which may be considered a potential source for human infection [20,21,22].

Our study revealed a similar seroprevalence for *N. caninum* in individual cattle samples (26.2%) to that detected by Hall et al. (2006), in Australia (21.1%) [55], higher than that found by Gerges et al. (2018) in upper Egypt (10%) [37], and lower than that reported by Enachescu et al. (2014) in Romania (45%) [53], all in individual cow’s milk samples. Additionally, our individual seroprevalence rate in sheep (33.3%) was markedly higher than reported by Al-Jomaily and Al-Rubaie, (2013) in Iraq (1.92%) [14] and lower than that recorded by Tamponi et al. (2015) in Italy (44%) [15]. 

Regarding *T. gondii* seroprevalence, our individual results in sheep (66.7%) were similar to that detected by Saad et al. (2018) in Upper Egypt (60%) [21] and higher than that of Mohamed et al. (2019) in Qena, Egypt (34%) [39]. In the case of goats, our individual results (81.8%) were higher than those of Gazzonis et al. (2019) in Italy (63.3%) [49], Liu et al. (2021) in China (9.7%) [56], and Mohamed et al. (2019) in Qena, Egypt (60%) [39], but lower than that recorded by Saad et al. (2018) in Upper Egypt (90%) [21]. Our result in cattle (2.4%) was markedly lower than that previously recorded in Qena (64%) (Mohamed et al., 2019) [39], one of our tested governorates. However, this might be influenced by the testing of IgM antibodies in the latter study compared to IgG in our study. Furthermore, variations in reported prevalences may be attributable to differences in location, timing, tested animals, and testing procedures.

For risk factor assessment, a comparison between animal species revealed a higher risk of *T. gondii* infection in sheep and goats than in buffaloes and cattle, but no influence of species was revealed on *N. caninum* infection. It is accepted that sheep and goats are more susceptible to *T. gondii* infection than cattle and buffaloes [1,2,17]. Only cattle samples were considered when analysing the influence of region, breed, and management system, as these factors did not vary for the other species. None of these factors was significantly influencing seropositivity neither for *T. gondii* nor for *N. caninum*. 

The findings of the current study suggest the suitability of bulk milk samples of different ruminant species in monitoring *N. caninum* and *T. gondii* antibodies at the herd level in Egypt. This was also supported by the good agreement between the results of the individual samples compared to the overall bulk milk result. However, as only few bulk milk samples of goats, sheep and buffaloes were included, further studies testing combined individual and bulk milk samples are required to validate our results. Additionally, testing of individual samples is still necessary when aiming at detecting infected individuals, e.g., for replacing *N. caninum*-infected cows [54]. 

## 5. Conclusions

We provide novel data on the milk-based prevalence and risk factor assessment of *N. caninum* and *T. gondii* among cattle, buffaloes, sheep and goats from different areas in Egypt. The utility of bulk milk samples for detecting *N. caninum* and *T. gondii* antibodies was confirmed for the first time in Egypt. Furthermore, we report the first detection of *N. caninum* DNA in cow’s milk in Egypt. The high prevalence of *N. caninum* antibodies in cattle suggests a high economic impact of this parasite in Egypt. Moreover, high seroprevalence and DNA detection of *T. gondii* in sheep’s and goat’s milk possibly threatens both human and animal health and should alert public health and veterinary authorities.

## Figures and Tables

**Figure 1 pathogens-11-01305-f001:**
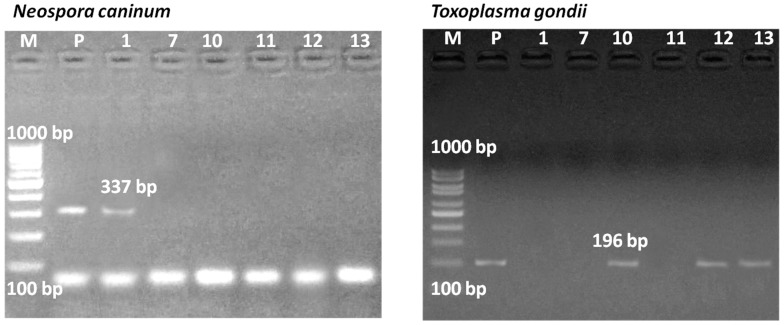
PCR results for *Neospora caninum* (left) and *Toxoplasma gondii* (right) DNA. Seropositive bulk milk samples and seronegative bulk milk samples containing a number of positive individual samples were analyzed by PCR (lanes 1, 7, 10, 11, 12, and 13, indicating the ID of the tested bulk milk samples). A fragment of *Nc5* gene (337 bp) of *N. caninum* and *B1* gene (196 bp) of *T. gondii* was amplified, and the product was run on 1.5% agarose gel. Bulk 1 was positive for *N. caninum*, and bulks 10, 12, and 13 were positive for *T. gondii*. Lane M: 100 bp DNA ladder. Lane P: positive control.

**Figure 2 pathogens-11-01305-f002:**
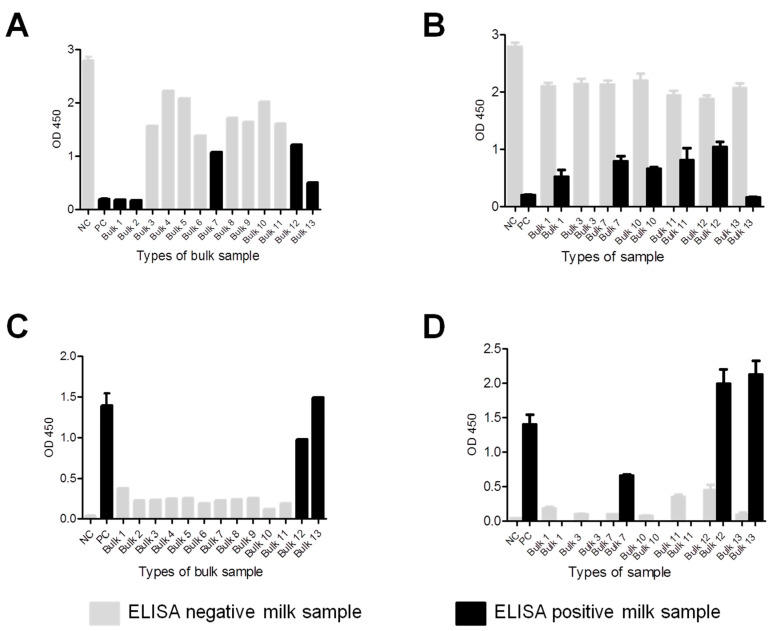
Antibody levels of bulk and individual milk samples to *Neospora caninum* and *Toxoplasma gondii.* (**A**) Optical density (OD) of anti-*N. caninum* antibody levels in tested bulk samples, negative (NC), and positive controls (PC). (**B**) Mean OD of *N. caninum* negative and positive individual milk samples in each bulk sample. (**C**) OD of anti-*T. gondii* antibody levels in tested bulk samples, negative (NC), and positive controls (PC). (**D**) Mean OD of *T. gondii* negative and positive individual milk samples in each bulk sample.

**Figure 3 pathogens-11-01305-f003:**
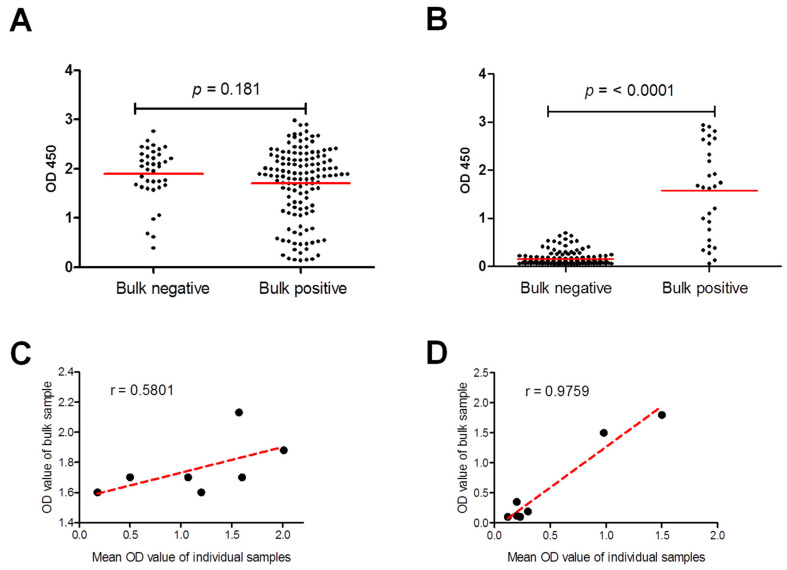
Comparison and correlation of antibody level of bulk and individual milk samples to *Neospora caninum* and *Toxoplasma gondii.* (**A**) Comparison of the antibody levels of all tested individual samples to those of bulk samples with negative and positive ELISA results for *N. caninum* (**A**) and *T. gondii* (**B**). Red lines represent mean values of optical density (OD) at 450. Statistical analyses were applied by Student *t*-test followed by Mann-Whitney test. (**B**) Pearson’s correlation coefficient of ELISA OD of bulk and mean OD of individual samples for *N. caninum* (**C**), and *T. gondii* (**D**). Scatter graphs show the correlation between absorbance values in the ELISA of bulk and individual samples. Correlation coefficients were calculated using Pearson’s correlation coefficient: |r| = 0.70, strong correlation; 0.5 < |r| < 0.7, moderately strong correlation; and |r| = 0.3–0.5 weak-to-moderate correlation. Correlation coefficient (r): r = 0.580 for *N. caninum*, and r = 0.9759 for *T. gondii*.

**Figure 4 pathogens-11-01305-f004:**
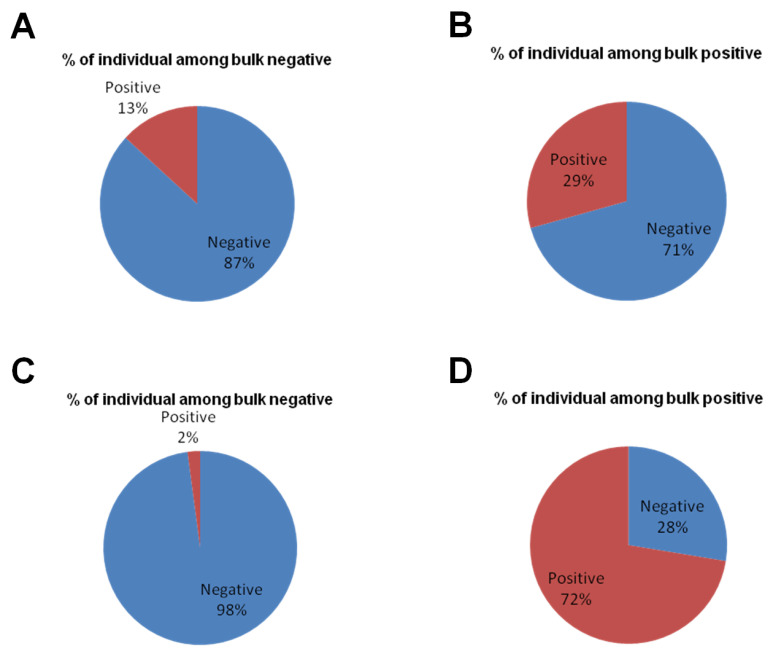
The positive/negative ratio of individual samples within negative and positive bulk samples. In case of *N. caninum*, the percentage (%) of positive individual samples was 13% (5/38) in the negative bulk groups (**A**); % of positive individual samples in positive bulk was 29% (39/133) (**B**). For *T. gondii*, % of positive individual samples within the negative bulk samples was 2% (3/142) (**C**), and % of positive samples in the positive bulk was 72% (21/29) (**D**).

**Table 1 pathogens-11-01305-t001:** Details of collected bulk and individual milk samples.

Group	Species	Breed	Region	Management *	Sampling Place **	Testing of Individual Samples (Number)
Bulk 1	Cattle	Holstein Friesian	Qena	Large farm	Farm	Yes (34)
Bulk 2	Cattle	Holstein Friesian	Qena	Large farm	Shop	No
Bulk 3	Cattle	Holstein Friesian	Qena	Small farm	Farm	Yes (12)
Bulk 4	Cattle	Holstein Friesian	Qena	Small farm	Shop	No
Bulk 5	Cattle	Native	Qena	Small holder	Shop	No
Bulk 6	Cattle	Holstein Friesian	Qena	Large farm	Shop	No
Bulk 7	Cattle	Holstein Friesian	Sohag	Large farm	Farm	Yes (70)
Bulk 8	Cattle	Holstein Friesian	Cairo	Small farm	Shop	No
Bulk 9	Cattle	Holstein Friesian	Cairo	Small farm	Shop	No
Bulk 10	Cattle	Native	Dakahlia	Small holder	Farm	Yes (10)
Bulk 11	Buffalo	Native	Dakahlia	Small holder	Farm	Yes (16)
Bulk 12	Sheep	Native	Dakahlia	Small holder	Farm	Yes (18)
Bulk 13	Goat	Native	Dakahlia	Small holder	Farm	Yes (11)

* large farm: >200 animals; small farm: 50–200 animals; small holder: <50 animals; ** Shops sell the daily milk yield of one farm either from a large container or aliquoted in plastic bags.

**Table 2 pathogens-11-01305-t002:** Target genes, primer sequences, amplicon sizes, cycling conditions and references of PCRs.

Target Gene	Primers Sequences	Amplified Segment (bp *)	Primary Denaturation (94 °C)	Amplification (35 Cycles)			Final Extension (72 °C)	Reference
				Secondary Denaturation (94 °C)	Annealing	Extension (72 °C)		
*Toxoplasma gondii B1*	5′-GGAACTGCATCCGTTCATGAG-3′	196	5 min	30 s	60 °C 30 s	30 s	7 min	[41]
	5′-TCT TTA AAG CGT TCG TGG TC-3′							
*Neospora caninum Nc5*	5′-CCCAGTGCGTCCAATCCTGTAAC-3′	337	5 min	30 s	63 °C 40 s	40 s	10 min	[42]
	5′-CTCGCCAGTCAACCTACGTCTTCT-3′							

* bp; base pairs.

**Table 3 pathogens-11-01305-t003:** Detection of antibodies to, and DNA of, *Neospora caninum* and *Toxoplasma gondii* in bulk milk samples.

Group	Species	*Neospora caninum*			*Toxoplasma gondii*			Co-Infection
		Sample to Negative Ratio (S/N%)	Interpretation	PCR Result	Sample to Positive Ratio (S/P%)	Interpretation	PCR Result	Interpretation
Bulk 1 (Qena)	Cattle	**6.7**	**pos.**	**pos.**	22	neg.	neg.	neg.
Bulk 2 (Qena)	Cattle	**6.5**	**pos.**	ND	12.4	neg.	ND	neg.
Bulk 3 (Qena)	Cattle	58.4	neg.	ND	12.5	neg.	ND	neg.
Bulk 4 (Qena)	Cattle	82.8	neg.	ND	13.6	neg.	ND	neg.
Bulk 5 (Qena)	Cattle	77.6	neg.	ND	14	neg.	ND	neg.
Bulk 6 (Qena)	Cattle	51.4	neg.	ND	10	neg.	ND	neg.
Bulk 7 (Sohag)	Cattle	**39.8**	**pos.**	neg.	12.2	neg.	neg.	neg.
Bulk 8 (Cairo)	Cattle	63.7	neg.	ND	13.1	neg.	ND	neg.
Bulk 9 (Cairo)	Cattle	61.1	neg.	ND	14.2	neg.	ND	neg.
Bulk 10 (Dakahlia)	Cattle	75.1	neg.	neg.	5.4	neg.	**pos.**	neg.
Bulk 11 (Dakahlia)	Buffalo	59.8	neg.	neg.	10.2	neg.	neg.	neg.
Bulk 12 (Dakahlia)	Sheep	**44.9**	**pos.**	neg.	**60.7**	**pos.**	**pos.**	**pos.**
Bulk 13 (Dakahlia)	Goat	**18.6**	**pos.**	neg.	**94**	**pos.**	**pos.**	**pos.**
Prevalence			5/13 (38.5%)	1/6 (16.7%)		2/13 (15.4%)	3/6 (50%)	2/13 (15.4%)

ND: not done; pos. = positive; neg. = negative.

**Table 4 pathogens-11-01305-t004:** Seroprevalence of *Neospora caninum*, *Toxoplasma gondii*, and mixed infection in individual milk samples of tested animals in Egypt.

Animal Species	No. of Tested	*N. caninum*			*T. gondii*			Co-Infection		
		No. of Negative (%)	No. of Positive (%)	95% CI *	No. of Negative (%)	No. of Positive (%)	95% CI *	No. of Negative (%)	No. of Positive (%)	95% CI *
Cattle	126	93 (73.8)	33 (26.2)	18.9–34.9	123	3 (2.4)	0.6–7.3	124 (98.4)	2 (1.6)	0.3–6.2
Buffalo	16	13 (81.2)	3 (18.8)	5–46.3	16 (100)	0	0–24.1	16 (100)	0	0–24.1
Sheep	18	12 (66.7)	6 (33.3)	14.3–58.8	6 (33.3)	12 (66.7)	41.2–85.6	15 (83.3)	3 (16.7)	4.4–42.3
Goat	11	9 (81.8)	2 (18.2)	3.2–52.2	2 (18.2)	9 (81.8)	47.8–96.8	10 (90.9)	1 (9.1)	0.5–42.9
Total	171	127 (74.3)	44 (25.7)	19.5–33.1	147 (86)	24 (14)	9.4–20.4	164 (95.9)	6 (3.5)	1.4–7.8

* 95% CI calculated according to method described by (http://vassarstats.net/), accessed on 11–14 September 2022.

**Table 5 pathogens-11-01305-t005:** Risk factors for *Neospora caninum* antibodies in milk samples in Egypt.

Analyzed Factor	No. of Tested	No. of Negative (%)	No. of Positive (%)	OR (95% CI) ^#^	Fisher Test ^x^
Region *					
Qena	46	37 (80.4)	9 (19.6)	Ref	Ref
Sohag	70	48 (68.6)	22 (31.4)	0.49 (0.2–1.2)	0.19
Dakahlia	10	8 (80)	2 (20)	0.97 (0.18–5.4)	1
Animal species					
Cattle	126	93 (73.8)	33 (26.2)	0.62 (0.13–3.1)	0.73
Buffalo	16	13 (81.3)	3 (18.7)	0.96 (0.13–7)	1
Sheep	18	12 (66.7)	6 (33.3)	0.44 (0.07–2.7)	0.67
Goat	11	9 (81.8)	2 (18.2)	Ref	Ref
Breed *					
Native	10	8 (80)	2 (20)	Ref	Ref
Holstein Friesian	116	85 (73.3)	31 (26.7)	0.69 (0.14–3.4)	0.73
Management system *					
Small holder	10	8 (80)	2 (20)	Ref	Ref
Farm (Small or large)	116	85 (73.3)	31 (26.7)	0.69 (0.14–3.4)	0.73

* Cattle only were assessed for these factors; ^#^ Odds ratio at 95% confidence interval as calculated by http://vassarstats.net/; ^x^
*p* value was evaluated by Fisher Exact Probability Test (two-tailed) using online statistics software http://vassarstats.net/, accessed on 11–14 September 2022, and GraphPad Prism version 5; Ref; value used as a reference.

**Table 6 pathogens-11-01305-t006:** Risk factors for *Toxoplasma gondii* antibodies in milk samples in Egypt.

Analyzed Factor	No. of Tested	No. of Negative (%)	No. of Positive (%)	OR (95% CI) ^#^	Fisher Test ^x^
Region *					
Qena	46	46 (100)	0	4.8 (0.24–95.6)	0.27
Sohag	70	67 (95.7)	3 (4.3)	Ref	Ref
Dakahlia	10	10 (100)	0	1.1 (0.05–22.6)	1
Animal species **					
Cattle	126	123 (97.6)	3 (2.4)	82 (18.2–370.2)	<0.0001
Buffalo	16	16 (100)	0	63.5 (3.3–1236)	<0.0001
Sheep	18	6 (33.3)	12 (66.7)	Ref	Ref
Goat	11	2 (18.2)	9 (81.8)	0.4 (0.07–2.7)	0.67
Breed *					
Native	10	10 (100)	0	0.65 (0.03–13.4)	1
Holstein Friesian	116	113 (97.4)	3 (2.6)	Ref	Ref
Management system *					
Small holder	10	10 (100)	0	0.65 (0.03–13.4)	1
Farm (Small or large)	116	113 (97.4)	3 (2.6)	Ref	Ref

* Cattle only were assessed for these factors; ^#^ Odds ratio at 95% confidence interval as calculated by http://vassarstats.net/; ^x^
*p* value was evaluated by Fisher Exact Probability Test (two-tailed) using online statistics software http://vassarstats.net/, accessed on 11–14 September 2022, and GraphPad Prism version 5; ** The result is significant at *p* < 0.0001; Ref; value used as a reference.

**Table 7 pathogens-11-01305-t007:** Comparison of bulk milk results with individual milk samples for antibodies to *Neospora caninum* and *Toxoplasma gondii*.

Group	Species	Number of Individual Samples Tested	*Neospora caninum*		*Toxoplasma gondii*	
			Result of Bulk Milk	Positive Individual Samples (%)	Result of Bulk Milk	Positive Individual Samples (%)
Bulk 1	Cattle	34	**pos.**	9 (26.5)	neg.	0
Bulk 3	Cattle	12	neg.	0	neg.	0
Bulk 7	Cattle	70	**pos.**	22 (31.4)	neg.	3 (4.3)
Bulk 10	Cattle	10	neg.	2 (20)	neg.	0
Bulk 11	Buffalo	16	neg.	3 (18.8)	neg.	0
Bulk 12	Sheep	18	**pos.**	6 (33.3)	**pos.**	12 (66.7)
Bulk 13	Goat	11	**pos.**	2 (18.2)	**pos.**	9 (81.8)
Total		171		44 (25.7)		24 (14.0)

## Data Availability

All data generated and analyzed during this study are included in this published article. Raw data supporting the findings of this study are available from the corresponding author on request.

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
