# Peer review of "Prevalence of Neospora caninum and Toxoplasma gondii Antibodies and DNA in Raw Milk of Various Ruminants in Egypt"

_pathogens, 2022, doi:10.3390/pathogens11111305_

Round 1

Reviewer 1 Report

In this manuscript, the authors aim to determine the prevalence of Neospora caninum and Toxoplasma gondii antibodies and DNA in Raw Milk of ruminants from Egypt.

I have the following comments:

1. Abstract and manuscript: please replace "antibodies against..." by “antibodies to...". Correct throughout the manuscript. Also take this suggestion into consideration when the term "against" is used in an isolation form.

2. Please rewrite the following sentence. English editing is necessary: “Regarding individual samples (n = 171), antibodies were detected in 25.7%, 14%, and 3.5% against N. caninum, T. gondii, and co-infection, respectively.” Similar phrases should also be edited throughout the manuscript.

3. Keywords: were the animals included in the study ill? The terms "toxoplasmosis" and "neosporosis" are not indicative of the health status of the animals included in the study. I suggest replacing them with: Neospora caninum and Toxoplasma gondii.

4. Suggestion: keywords in alphabetical order.

5. Line 41: please add "sporulated" to read as "ingestion of sporulated oocysts".

6. Line 54: "protozoan", please correct.

7. Lines 56, 57: Please rephrase the sentence. The way it is written it implies that the tachyzoite is a frequent source of oral infection, which is not true. This stage of development is very sensitive to external environmental conditions and gastric enzymes.

8. Materials and methods: what were the criteria for the selection of these geographical regions and the farms for testing individual samples?

9. Materials and methods: How were the sample size calculated in both cases: bulk raw milk and individual samples? Was sampling carried out at random? Give more information.

10. Tables 3 and 7: The interpretation of the symbols in the Tables gets confusing. Please replace, for example, Pos. and Neg.

11. In point 3.2 (Risk factors assessment...), I suggest that the authors only put information in text about significant results. Non-significant differences are already described in Table 5, so there is no need to repeat the information.

12. Discussion: In my opinion the Tables presented can be suppressed. Authors should focus on discussing the results obtained in their study and make plausible comparisons by discussing the results obtained in the present study with similar ones. This is not a manuscript intended as a review of previous studies.

Reviewer 2 Report

This study investigated the prevalence of Neospora caninum and Toxoplasma gondii in various Ruminants in Egypt

Mayor comments:

1)   the first paragraph of the discussion section would be an introduction, but in the second paragraph there is no longer a discussion of the results.

2)      delete tables 8 and 9, they are not used in a discussion.

 Minor comments:

L107: how was the standardisation of this kit for serum and plasma to milk samples?

L219: describe the risk factors in the materials and methods section

Reviewer 3 Report

A widespread parasitic disease causing mass abortions in cows. It is known that 12-45% of aborted fetuses of dairy cattle are infected with N. caninum.    The article is well written, divided into logical and interrelated sections. Materials and methods are described in full.   I would like to complete the article with the following information: 1) Was Ncaninum cultured in nutrient media? 2) Age of aborted fetuses 3) The season of abortions in positively responding animals 4) The source of spread of the infection.

Round 2

Reviewer 2 Report

The authors have responded to each of the reviewers´ comments and suggestions.